# Baboons, Centipedes, and Lemurs: Becoming-Animal from *Queer* to *Ghost of Chance*

## Alexander Greiffenstern

Independent Scholar, Lehener Straße 59, 79106 Freiburg, Germany; Alexander.Greiffenstern@uni-due.de

**Abstract:** The paper establishes a connection between the becoming-writer of Burroughs, who found his calling and style during the 1950s and his signature characteristic of becoming-animal. This can first be observed in *Queer*, where Burroughs develops his so-called routine; a short sketch-like text that often involves instances of metamorphosis or transformation. The theoretical background for this short form and the term becoming-animal is taken from Deleuze's and Guattari's book on Kafka, who also worked best in short texts and frequently wrote about animals. "The Composite City" may be the central text to understanding Burroughs' work. It is the text where Burroughs found his style and his identity as a writer. Becoming-animal is a logical consequence that further develops Burroughs' aesthetic ideal. Over the following decades, he experimented with it in different forms, and toward the end of his career, it became part of an environmental turn. In *Ghost of Chance*, one can find the same aesthetic ideal that starts Burroughs' writing in 1953, but the political implications have turned toward saving the lemurs of Madagascar.

**Keywords:** William S. Burroughs; *Queer*; *Ghost of Chance*; *Yage Letters*; *Naked Lunch*; Madagascar

"As a child I had been a great dreamer, bordering on hallucinations which often involved animals. After years of trying to discover who and what I was, I suddenly awoke one morning and realized I didn't care. I didn't want insight. I wanted to escape and forget". (William Burroughs in Bauer (1981), p. 506)

The texts of William S. Burroughs are populated by all kinds of different animals. There are, among others, baboons, cats, lemurs, and centipedes. Especially in his late texts, animals play an important role for self-reflection, and their deep connections to humans are at the center of two slim but interesting books: *The Cat Inside* (1992) and *Ghost of Chance* (1991). While *The Cat Inside* is a peculiar memorial to the cats that lived with Burroughs during his last years in Lawrence, Kansas, *Ghost of Chance* is clearly connected to Burroughs' beginning as a writer and his aesthetic development. Chad Weidner remarks in his introduction to *The Green Ghost* about *Ghost of Chance*: "The novella has received virtually no scholarly attention, but it can reveal much about the writer's late varied use of narrative strategies to convey the need for urgent ecological restoration" (Weidner 2016, p. 20). On a topical level, the book can be read as a parable on modern society based in 18th century Madagascar and a fictional story about its lemurs. However, on an aesthetic level, the text illustrates the concepts of becoming-animal, as developed by Gilles Deleuze and Felix Guattari, that has been present in Burroughs' texts since the early 1950s. Burroughs himself frequently pointed to Kafka as one of his literary influences, and the short form seems a good indicator that connects the two authors. In 1997, Timothy Murphy analyzed Burroughs' work on the basis of Deleuze's philosophy, but he mentions Kafka only once in connection to *Nova Express* (p. 133), and becoming-animal plays no role at all because his focus is on other aspects of Deleuze's work. Nevertheless, "understanding Kafka is such a convenient key to understanding Burroughs" (Meyer 1990, p. 226) that a closer look at Burroughs' animals should start with Kafka.

## 1. Becoming-Animal in the Texts of Kafka

The beginning of Kafka's story "The Metamorphosis" is one of the most famous openings: "As Gregor Samsa awoke one morning from uneasy dreams he found himself transformed in his bed into a gigantic insect" (Kafka 1948, p. 67). In the course of the story, it remains unclear whether the metamorphosis of the title refers to the change in Gregor or the change of his family and their attitudes toward Gregor and their own lives. In the end, the change of the family is successful for the parents and Gregor's sister: for them, it is a new beginning. Gregor's becoming-animal somewhat fails. He is unable to flee the obligations of his job and of family life. He dies in his insect form without helping his family or himself.

In *Kafka: Toward a Minor Literature*, Deleuze and Guattari offer a reading of Kafka's texts. They defend Kafka against a critique from the political left that regarded Kafka as a bourgeois author and instead they show his radical artistic perspective. They argue that the struggle of a minority within a dominant culture becomes visible in their language and literature. Kafka as a German-speaking Jew in Czech Prague can be used as a blueprint for what they call "minor literature". The act of becoming-animal is an escape route from the ruling culture.

"To the inhumanness of the 'diabolical powers', there is the answer of a becoming-animal: to become a beetle, to become a dog, to become an ape, 'head over heels and away', rather than lowering one's head and remaining a bureaucrat, inspector, judge, or judged". (Deleuze and Guattari [1975] 1986, p. 12).[1]

Deleuze and Guattari regard this becoming-animal as a movement of deterritorialization, even as absolute deterritorialization (pp. 13, 36), which means that the becoming-animal in Kafka's stories has the aim to leave the symbolic order in which they live. The problematic struggle that follows a transformation is often presented in the story. "The Metamorphosis" has already taken place when the story starts, which is then about Gregor trying to live in his new body, but his old cultural context continues to reterritorialize him via his family and his job and eventually kills him.

Becoming-animal is always a threat to society, because it challenges established rules and produces lines of flight. There is no romantic connotation to becoming-animal as in common back-to-nature movements. The animals simply submit to other rules as in "Josephine". It is of importance that Kafka seems fascinated with small animals such as mice or bugs, because the transformation is perceived as dangerous, but the smaller life-form is usually not. The society of mice or a single bug can exist within the dominant culture and even use it as a host similar to a parasite or virus, which is the appeal that the small animals have to the author of minor literature. An important "characteristic of minor literatures is that everything in them is political" (Deleuze and Guattari [1975] 1986, p. 17), and in their later work, *A Thousand Plateaus*, Deleuze and Guattari elaborated on becoming-animal that minoritarian, as in minor literature, is not to be confused with a minority. "The majority in the universe assumes as pregiven the right and power of man. In this sense women, children, but also animals, plants, and molecules, are minoritarian" (Deleuze and Guattari [1980] 1987, p. 291).

Kafka himself stated that he wanted to avoid metaphors and an interpretation of Kafka's texts based on an analysis of metaphors is bound to fail. Therefore, Deleuze and Guattari see the metamorphosis as contrary to the metaphor and try to analyze the transformations and read animals "realistically". This leads to a different perspective: if it is of no interest what a bug stands for, the movement or transformation will become important. What drives this transformation? How is one territory connected to the other? Does the deterritorialization lead to destruction or to further transformation? Above all: How does the machine function? No story of Kafka embodies this question better than "The Penal Colony", which is misquoted in one of Burroughs' letters and is the source of "Dream of the Penal Colony", a short story/routine that appears in William S. Burroughs'

---

1   Quotes taken from a letter to Max Brod.

*Queer*. Kafka's short story is about a writing machine, and in a way, *Queer* shows the becoming-writer of Burroughs.

## 2. Becoming-Animal in *Queer*

As Oliver Harris has pointed out, to understand the becoming-writer of Burroughs and to grasp the development that resulted in the writing of *Naked Lunch*, one has to look at *Queer*, which is even the origin of the very title *Naked Lunch*. In his 2003 study of Burroughs' early work until *Naked Lunch*, Harris states that "*Queer* is the true kernel of *Naked Lunch*" (Harris 2003, p. 38). In the introduction to the new edition in 2010, Harris writes: "The origins of *Naked Lunch* in the cracking up of *Queer* appear quite literally at the moment when Lee first attempts to court Allerton with a friendly greeting and 'there emerged instead a leer of naked lust, wrenched in the pain and hate of his deprived body and, in simultaneous double exposure, a sweet child's smile of liking and trust, shockingly out of time and place, mutilated and hopeless'" (Burroughs [1985] 2010, p. xviii).

Burroughs wrote *Queer* in 1952, but it was not published until 1985, and afterwards, it was overlooked by critics for several reasons. The book was still a fragment, and Burroughs took the last stage of the manuscript and added an epilogue and an introduction that pointed readers into the direction of the shooting of his wife Joan of which there is no textual proof in *Queer*. Many readers expected to read something akin to an earlier version of *Naked Lunch* and finally discover the missing link between *Junky* and his masterpiece. Since this was not as straightforward as was expected, the pre-publication hype for the book died away quickly. The book has two aspects that seem important when it comes to transformations: first, William Lee's coming to terms with his homosexuality and his love and/or desire for Eugene Allerton; second, the usage of routines, which is a narrative form created by Burroughs that basically constitutes all of *Naked Lunch*, which Lee deploys to seduce Allerton, the object of his "naked lust".

While Lee's struggle with his homosexuality is the theme of the book, it appears first centrally in the opening scene of chapter 8 when Lee and Allerton are in Guayaquil, Ecuador. The chapter starts with a description of the city that foreshadows "The Composite City", which Burroughs sent in a letter to Ginsberg on 10 July 1953. Whereas "The Composite City" is a visionary or even utopian text, Guayaquil is "realistically" described through the eyes of Lee. This difference is also seen a few lines further down when Lee/Burroughs reflects about his chosen image.

"What happens when there is no limit? What is the fate of The Land Where Anything Goes? Men changing into huge centipedes . . . centipedes besieging the houses . . . a man tied to a couch and a centipede ten feet long rearing up over him. Is this literal? Did some hideous metamorphosis occur? What is the meaning of the centipede symbol"? (Burroughs [1985] 2010, p. 92).

The answer is that there is no fixed meaning. Probably this became clearer to Burroughs in the following years, but his centipede and other animals are not symbols; becoming-animal emerged as part of his writing similar to Kafka. As Kafka's animals, Burroughs' are not metaphorical animals, the writing is not allegorical, but always literal. For both authors, becoming-animal offers a way out. In his comparison of the two authors, Adam Meyer points out: "Kafka and Burroughs use images of metamorphosis to show this sort of death-in-life existence, but these portraits can also serve a second, opposite purpose: to show the man who, however quixotically, dares to fight against the system" (Meyer 1990, p. 220). For Burroughs, these metamorphoses are associated with corporeal questions of identity.

For Lee, in *Queer*, the image of a centipede starts a transformation into an animal-creature, some kind of predator searching for an object of desire. Lee's metamorphosis is signaled in the next paragraph: "The river looked as if a nameless monster might rise from the green-brown water. Lee saw a lizard two feet long run up the opposite bank" (p. 92). After he finds a group of young boys at the waterfront, he "looked at them openly, a cold stare of naked lust. He felt the tearing ache of limitless desire" (p. 93). Similar to

any animal on the hunt, he singles out one of the boys as his target. "The boy vibrated with life like a young animal" (p. 93). Then, he has a vision of being inside the boy's body experiencing memories of sexual encounters with the other boys and also with a woman. At this point, he snaps out of the vision and asks himself again if he is homosexual. "'I'm not queer,' he thought. 'I'm disembodied'" (p. 94). The scene ends rather comically with Lee asking Allerton "'Think I'm queer or something?' 'Frankly, yes'" (p. 95).

As Deleuze and Guattari suggest for Kafka, the becoming-animal is a line of flight. By becoming a predatory lizard on the river bank, Lee is able to express his desires (as scary as they might be) in contrast to his relation to Allerton in which he is afraid of being rejected as "some old queer". The metamorphosis is introduced by the idea of the centipede and then triggered by the lizard. It does not matter into what animal Lee changes, because the animals are not symbolic; it is a literal transformation, a becoming-animal as such. Lee's identity is not fixed; he is a homosexual but not a "fag". Despite being homosexual, he is the typical ugly American who behaves similar to a colonizer on several occasions, and in consequence, he is also an anti-communist, challenging a common point of view of the 1950s that equated homosexuality and communism as similar threats undermining American society. However, this is only one possible reading, because the ugly American can also be some kind of act leading up to his numerous routines, which often go in a similar direction (such as the Slave Trader or the Texas Oil Man). Then, there is the sentence Lee says to Allerton, "Just a routine for your amusement, containing a modicum of truth" (p. 99), pointing to the meta-level of the text that a lot of the routines were probably written to amuse real-life Allerton Lewis Marker. So, the adopting of roles and making up of the routines can be seen as a movement toward a somehow fixed identity. In the other direction, becoming-animal and taking up new roles represent the way out of his struggle, which is exactly what this fixed identity is. For Burroughs, the consequence is that William Lee becomes an assemblage where identity as such plays no role anymore; how the mechanisms within the assemblage are connected matters instead. The imagery for this aesthetic, as it can later be observed in *Naked Lunch*, is already at work in *Queer*.

### 3. The Yage Letters

One central text to understand Burroughs' work and his becoming as a writer is surely "The Composite City" written in 1953 and intended for *The Yage Letters* (1963). The books consist of three parts: The main part "In Search of Yage" describes in epistolary form Burroughs' travel through South America where he searched for the mythical drug yagé. In *The Yage Letters*, it becomes increasingly difficult to differentiate between Burroughs' alter ego William Lee and the author himself, because some letters are real letters that Burroughs sent to Allen Ginsberg, while others are not. All letters are addressed to Ginsberg, but they are signed differently. The second part of the book is a short exchange of letters between Burroughs and Ginsberg called "Seven Years Later (1960)", and the third called "Epilogue (1963)" consists of a note by Ginsberg and a cut-up text by Burroughs, signed "William S. Burroughs". "The Composite City" is the last letter of the main part and seemingly the result of Burroughs' quest; it is also part of a real letter to Ginsberg. The introduction of "The Composite City" signifies foremost a change of perspective. For the protagonist/author, a movement has occurred that is summarized in the first sentence of the vision: "Yage is space time travel" (Burroughs and Ginsberg (2006), p. 50). With this first sentence, Burroughs does not engage with the genres of science-fiction or fantasy—something that will happen later in his work—it is meant with metaphysical considerations. It means that every period of time and every place on earth has left some kind of mark on the body of the individual.

As for how far the vision of the Composite City was a result of Burroughs' consumption of yagé, or rather of his artistic considerations while traveling alone through South America, it does not matter in the end. With his first sentence, Burroughs signals that he experienced a change in perspective and describes how he felt in touch with every(-)body:

"Yage is space time travel. The room seems to shake and vibrate with motion. The blood and substance of many races, Negro, Polynesian, Mountain Mongol, Desert Nomad, Polyglot Near East, Indian—new races as yet unconceived and unborn, combinations not yet realized passes through your body. Migrations, incredible journeys through deserts and jungles and mountains (stasis and death in closed mountain valleys where plants sprout out of your cock and vast crustaceans hatch inside and break the shell of the body), across the Pacific in an outrigger canoe to Easter Island. The Composite City where all human potentials are spread out in a vast silent market" (p. 50).

The journey to the Composite City leads through the body, and it mirrors the races and migrations of the earth. Again, apart from describing a transcendental experience, he describes movements and transformations. Burroughs is not moving; everything around him "shakes and vibrates with motion". Nevertheless, he has to change too—while the world in all four dimensions circulates around him and through him, Burroughs becomes something else.

Burroughs becomes a writer who incorporates the movement of the world into his writing. He denounces his nationality; being a U.S. citizen does not matter in the Composite City. It seems as though he does not want to be the Ugly American anymore, who the Lee of *Queer* so perfectly portrays. He wants to become a citizen of the singular cities such as New York City, Lima, London, Paris, or Tangier. In this sense, the Composite City has the function of a utopian society in which race is of no concern. For a white Caucasian from a wealthy mid-western family who graduated from Harvard, it was pretty bold to claim that "Negro" blood passes through his body, even if it just for the duration of his vision.

A refusal of American politics is the routine "Roosevelt after Inauguration" that is also part of the original "In Search of Yage" manuscript. As a result of its offensive character, the routine was excluded from the text until the third edition in 1988.[2] This routine has all the typical ingredients of a typical Burroughs routine. It starts with the simple idea that the president can theoretically appoint whoever he likes for his cabinet and that these politicians are basically crooks. Then, this idea is combined with the question of whether there is a difference between politicians and small-time thieves who rob drunks in the subway. The answer is of course that they have a very different appearance and mode of operation, but apart from that, they are basically the same. So, in the routine, Roosevelt appoints people from the lower ranks of society to the highest offices in government, thereby leveling the usual hierarchy. The routine is politically incorrect and funny when it lists the shady characters and their positions. "*Attorney General*: A character known as 'The Mink', a peddler of unlicensed condoms and short-con artist" (p. 42). After that, the Supreme Court gets involved, and the routine becomes offensive. Roosevelt and his advisor Harry Hopkins force the justices to have intercourse with baboons, and one by one, the members of the Supreme Court are replaced by the simians. Then, he kills or disempowers the members of congress and senate to install a system that has the aim to dehumanize every citizen. The routine ends with Roosevelt saying, "I'll make the cocksuckers glad to mutate" (p. 45).

The routine presents no human-to-animal transformation as such, but it raises some interesting questions and shows Burroughs' early usage of animals in his writing. Roosevelt was already dead in 1953, and his merits are without question. Why Burroughs chose Roosevelt and not Truman or Eisenhower can be an indication that at least the idea for this routine stems from the mid 1940s when Burroughs was still in New York City or trying his luck as a respectable farmer in South Texas. From Burroughs' point of view, the "New Deal" certainly interfered with his idea of free enterprise and could only be regarded as communism (cf. Johnson 2006, pp. 98–9). The whole joke is based on the notion that politicians often act self-serving and follow their instincts—good political instinct is even used as form of praise—rather than some ideal to serve the common good. In this sense,

---

2  City Lights publisher Lawrence Ferlinghetti was not happy with the routine, but it "would stay in right until the long galley stage, when the British distributors, fearing legal action, pressured the London-based printers, Villiers, to have it pulled" (Burroughs and Ginsberg xli). The routine was first published in 1964 as a mimeo pamphlet by Ed Sanders' Fuck You Press.

there is not much difference between a primate such as a baboon and a politician. For Burroughs, judges were probably on the same level as politicians. Nevertheless, Roosevelt did stack the Supreme Court, so he could secure his New Deal legislation, but, as usual, one is inclined to say that Burroughs' routine gets totally out of hand, and he has readable fun wreaking havoc and creating chaos. The last sentence is presented as the only way out, and at the same time, the consequence intended by Burroughs: mutation. Similar to the utopian composite city, transformation, becoming something else, offers a way out.

In the political routine "Roosevelt after Inauguration", the primates are threatening. The baboons represent the worst character traits of humans, and they take the places of human judges. Interestingly, it is the centipede that Burroughs returns to again and again to explore the power of becoming-animal.

### 4. The Centipede in *Naked Lunch*

The centipede appears in *Naked Lunch* on several occasions. Most notable here is the routine "Meeting of International Conference of Technological Psychiatry" because it contains a transformation of a man into a centipede. In this routine, "Doctor 'Fingers' Schafer, the Lobotomy Kid" has removed the brain of a man to create his "Master work", the so-called "*Complete All American Deanxietized Man*" (Burroughs 2009, p. 87) who transforms into a giant black centipede in front of the eyes of the conference, thereby turning against his creator, when at the end of the routine, the monster seemingly starts to attack the members of the conference. The routine stands in the literary tradition of the mad scientist such as Dr. Frankenstein or his descendants from the pulp magazines of the 1920s and 1930s. Yet, Burroughs is not just interested in the modern Prometheus or the horror of modern science; he points out a whole array of fears and monsters in American society.

Dr. Schafer wants to control his experimental humans by turning them into mindless "Drones" (p. 88). However, as one member of the conference points out when he sees the centipede, "Schafer has gone a bit too far" (p. 87). By removing his brain and creating a man free of all anxieties, he has also removed the constraints that held him in check. The mindless idiots he creates have only one way out, transforming into something completely different but somehow related to their current existence: invertebrates. In this case, the transformation is into a centipede, in the case of Gregor Samsa into some kind of bug. Of course, this anxiety-free creature is perceived as an instant threat to society by the members of the conference, because it has nothing to lose. It is not afraid to lose a job or how to repay a debt to the bank. So, the reaction shows the real "monster" of American society, racism.[3] "We must stop the Un-American crittah", says a fat frog-faced Southern doctor [ . . . ] "Fetch gasoline!" he bellows. "We gotta burn the son of a bitch like an uppity Nigra!" (p. 88). After realizing the "mother fucker's hungry", the conference members start panicking and rushing out. The routine ends at this point and does not answer whether the monster is really dangerous and about to attack somebody, or whether the transformation itself and the possibilities it represents are the real threat.

Burroughs adds another layer to the routine by interrupting the scene between the call for gasoline and the panic with a fantasy by a "cool hip young doctor high on LSD25" about a future trial after the killing of the centipede. Burroughs calls into question the difference between the law and a moral code. In the fantasy of the young doctor, "a smart D.A. could" be able to use the law for persecuting the ethical crimes committed by Doctor Schafer, because the killing of the centipede is perceived as murder of the human being before the transformation, for which there would be no evidence after a burning of the creature. The young doctor does not want to get involved into this possibility, "I'm not sticking my neck out", because he is afraid of a clever D.A. and not because he thinks they are killing a human being or a living creature that should not be harmed. This question

---

3   In his paper "The Dark Ecology of *Naked Lunch*", George Hart uses this scene to make the connection to speciesism. "In speciesism, humans must clearly be distinguished from centipedes just as, in racism, white people must be distinguished from Black people. Once the ambiguity is resolved, the violence is justified" (Hart 2020, p. 6).

comes to the foreground again when Burroughs returns to other primates in his late *Ghost of Chance*.

## 5. Ghost of Chance

A first look at Burroughs' fiction seems to suggest that there exist only two types of animals in his books: insects and wild mindless creatures such as the baboons in "Roosevelt After Inauguration". Nonetheless, the examples of the centipede in *Queer* and *Naked Lunch* show that becoming-animal is far more complex than that. In *Ghost of Chance*, a slim novel published in 1991, Burroughs sides with the lemurs of Madagascar to create another visionary place. The lemurs in *Ghost of Chance* can be seen in direct opposition to the baboons in the Roosevelt routine. While the baboons can only be perceived as wild creatures that rape the justices of the Supreme Court and kill them one by one, the lemurs have a personality and cultural significance. The becoming-animal in *Ghost of Chance* is connected to the idea of a utopian society.

The book consists roughly of two parts. One tells the story of Captain Mission who lives on Madagascar in the early 18th century. The second part describes the revenge of nature against mankind in the form of diseases. Captain Mission is a figure who also appears in Burroughs' *Cities of the Red Night* (1981), and it is not entirely clear whether he is solely a fictional character invented in the 18th century or whether the stories surrounding him are based on true accounts.[4] The story goes that Mission was a pirate captain who founded a free republic for pirates and freed slaves at the coast of Madagascar, which was called Libertatia. The governmental rules in Libertatia can be regarded as forerunners of the French Revolution, as every individual had the same rights.

In any case, Burroughs saw Libertatia as an attempt of people to free themselves from tyrannical governmental structures. So, in this respect, it is very much in line with his "critique" of the Roosevelt administration. The narrator sums up the articles of the democratic state, which Mission wrote together with the English captain Thomas Tew:

"The Articles were based on ideas remarkably like the ideas behind the French and American revolution of the late eighteenth century—and preceded them by more than sixty years. There would be no capital punishment, no slavery, no imprisonment for debt, and no interference with religion or sexuality" (Burroughs [1991] 2002, p. 57).

In *Ghost of Chance*, Captain Mission is the wise leader because he is in touch with his environment and especially with the famous lemurs of Madagascar. The story is a parable about how human civilization is seemingly moving forward by applying democratic structures and human rights, but that betrayal and violence are still present and the innocent victim of conflicts and greed is nature. In Burroughs' story, a lemur is killed by a certain Bradley Martin who wants to ignite a violent conflict between the settlers and the natives who regard the lemurs as sacred. Similar to Captain Mission, Bradley Martin is also a figure that appeared before in Burroughs' text, but with a much longer history. The figure is a product of Burroughs' cut-up project and the early manuscript of *The Soft Machine* (1961) was in fact titled "Mr Bradley Mr Martin" (cf. Burroughs 2012, p. 25). From there on, he makes appearances in many of Burroughs' texts, especially in the 1960s.

In *Ghost of Chance*, Martin is an agent of a secret society in London that wants to end the free republic but also regards Mission's behavior toward the lemurs as dangerous.

"A woman leans slightly forward. . . . She speaks in a cold, brittle voice, each word a chip of obsidian: 'There is a more significant danger. I refer to Captain Mission's unwholesome concern with *lemurs*.' The word slithers out of her mouth writhing with hatred" (8).

Captain Mission begins an intimate relationship with one lemur whom he calls Ghost. He starts living with the lemurs in an ancient stone structure, which suggests some connection of Madagascar to Atlantis or Mu. The becoming-animal that Mission willfully

---

4  For a discussion of Captain Mission in *Cities of the Red Night* and its source text, *The History of the Pyrates* (1724) by Daniel Defoe, cf. Robinson 2011, pp. 135–40.

starts is accompanied by a form of time travel. Here, Burroughs uses genre and steps into Lovecraft territory. Mission finds a book, *The Ghost Lemurs of Madagascar*, that tells the ancient history of the Lemur People that existed before humanity—similar to Lovecraft's Great Old Ones.[5] From the book, Mission learns about the history of the Lemur People, but Mission can not finish his transformation into one himself. He is tricked into leaving the island, and when he returns, the stone structure is destroyed, and Ghost is dead. Then, Mission curses the rest of the earth. After his own death, he seems to become a ghost, finishing the transformation into this special kind of lemur and haunting the so-called "Board" that is responsible.

As the summary already shows, *Ghost of Chance* is interesting for a lot of reasons. Burroughs connects the text to his earlier work, in particular "The Composite City". Before Mission discovers the ancient stone structure, he takes a local drug called "indri". The analogy to yagé and therefore to *The Yage Letters* is made in the text. "Mission had smoked opium and hashish and had used a drug called *yagé* by the Indians of South America. There must, he decided, be a special drug peculiar to this huge island, where there were so many creatures and plants not found anywhere else" (p. 10). It becomes clear in the second sentence that the line between Burroughs, the narrator, and Captain Mission is willfully thin, which connects the narrator to Burroughs' alter ego William Lee, the protagonist of *Queer* and *The Yage Letters*. "The drug was called *indri*, which meant 'look there' in the native language". The translation is right, but "indri" usually refers to a special species of lemurs that are also known as "babakoto", which means "ancestor of man". The lemur Ghost seems to belong to this species, which is taller and has much more similarities with humans than other species of lemurs. So, in a similar way that Burroughs sets off his vision of the Composite City by taking yagé, Captain Mission takes indri before embarking on his trip to the jungle and finding the ancient structure where Ghost lives.

Mission asks the native about the drug: "Is this a day or a night drug?" "Best at dawn and twilight" (p. 10). This shows the problems of binary thinking very graphically. As yagé, indri signifies the change of perspective; it is a drug of the different shades of gray. As Lee in *The Yage Letters*, Mission gets in touch with the ancestors—signified by the "Ancestor of Man" Ghost and the ancient stone structure—and after he has become a ghost himself, he travels into the future to witness the apocalypse.

During the apocalypse, we encounter an image taken from the Composite City: the plant growing from the penis. In *Ghost of Chance*, the apocalypse comes in the form of strange diseases that are all based on some form of becoming.

"Ravenous diseases lurk in dust and straw, mist and swamps and fossilized rock. Some of the deadliest are parasite plants specialized to grow in human flesh, like the Roots. Roots grow down into the viscera and glands, curling around bones; vines sprout from the victim's groin and armpits; *green shoots spring from his penis tip* [my emphasis]; tendrils creep out of his nostrils to release deadly seeds that then spread on the wind; thorns tear out his eyes; his testicles swell and burst with roots; his skull becomes a flowerpot for stunning brain orchids that grow over dead eyes and idiot face while the skin slowly toughens into bark. In some cases, metamorphosis is complete. The subject grows into the ground to know the exquisite agony of quickening sap, of leaves eating light and root nourished by water, shit, and soil". (p. 47).

In the Composite City, there is a parenthesis that says "stasis and death in closed mountain valleys where plants sprout out of your cock and vast crustaceans hatch inside and break the shell of the body" (*The Yage Letters* p. 50).[6] The becoming-plant is seeded in the Composite City. Whereas in the Composite City, the plants sprouting from a penis are a comical sign of stasis, in *Ghost of Chance*, the image has an environmental function that warns against the revenge of nature, which will inevitably happen because humans

---

5    In the essay collection *Retaking the Universe: William S. Burroughs in the Age of Globalization* (Schneiderman and Walsh), one finds an essay by "Ccru" called "Lemurian Time War" that explores the connection between Burroughs and Lovecraft by starting out from "The Ghost Lemurs of Madagascar" (1987), which is a slightly different and shorter version of *Ghost of Chance* (Ccru 2004).

6    In *Naked Lunch* "cocks" is substituted with "genitals".

are part of nature, too. The same way night and day are not opposites, human beings and animals and even plants are not opposites, either. Burroughs creates a disease that shows images of the twilight between humans and plants. The apocalypse cleanses the earth and "Green People" repopulate it. Burroughs' vision for the future is one of ecological harmony:

"The Four Horsemen ride through ruined cities and neglected, weed-grown farms. The virus is burning itself out, its victims dying by the millions. People of the world are at last returning to their source in spirit, back to the little lemur people of the trees and the leaves, the streams, the rocks, and the sky. Soon, all sign, all memory of the wars and the Plague of Mad will fade like dream traces" (p. 54).

In a short "Afterword", the narrator returns to Captain Tew and the end of the colony of pirates on Madagascar. For Burroughs—this time, it is clear that it is Burroughs and not some narrator—the loss of the Free Republic of Libertatia is clearly connected to the lemurs of Madagascar, because the texts ends with a footnote where Burroughs describes the situation of the lemurs today and asks for a donation to the Duke University Primate Center. Without a doubt, empathy is the first step toward understanding becoming-animal.

In 1953, Burroughs wrote "In Search of Yage", which is a book that shows Burroughs' fondness of movement and a book that is central for showcasing Burroughs' formation as a writer. This seems also strongly connected to a rather traditional question of identity. The answer to the question that is implicitly asked in the titles of Burroughs' first two novels, *Junky* and *Queer*, seems to be neither, or *Naked Lunch*. If *Naked Lunch* is some sort of answer to this search for an identity, this means that the concept of identity was replaced with an ever-evolving idea of becoming, with his becoming-writer the most important step that could be observed in his early texts. Burroughs incorporated becoming-animal into his works and embodied it himself as a writer who explored the edges of his art form with the development of the cut-up technique in the 1960s.

"The Composite City", a text written for "In Search of Yage", but first published in *Naked Lunch*, may be the central text to understanding Burroughs' work. It represents movement and the multitude of possibilities. It is the text where Burroughs found his style and his identity as a writer. Becoming-animal is a logical consequence that further develops Burroughs' aesthetic ideal. Over the following decades, he experimented with it in different forms, and toward the end of his career, it became part of an environmental turn. In 1991, in *Ghost of Chance*, one can find the same aesthetic ideal that starts Burroughs' writing in 1953, but the political implications have turned toward saving the lemurs of Madagascar.

Becoming-animal as defined by Deleuze and Guattari gives a terminology and aesthetic concept to describe Burroughs' writing. Furthermore, it connects him to a modernist writer such as Kafka, and his routines, which first became a central element in *Queer*, also make him comparable to Kafka's short fiction. Burroughs is clearly part of this modernist tradition, even though he and other writers give it their own spin, which made Timothy Murphy come up with the term "amodern" (cf. Murphy 1997, especially pp. 29–34) in contrast to postmodern.

It is astounding that a green book that reads is parts as a pamphlet by an animal-rights activist stands at the end of Burroughs' 40-year-long career as a writer. Then again, the similarities to his yagé vision of the Composite City are very visible, and Burroughs makes references to his other works as well. He refers to Brion Gysin and their cut-ups, and there is the connection to *Cities of Red Night* through Captain Mission. Furthermore, *Ghost of Chance* is a statement about his artistic development. In the book, one finds the text accompanied by abstract pictures that Burroughs had painted. By presenting texts from 1953 to 1963, *The Yage Letters* showed the development of becoming a montage-writer and a decade later the cut-up artist. *Ghost of Chance* refers back to his early texts and the concept of becoming-animal and the closely related becoming-plant. Throughout his texts, becoming-animal offered Burroughs possibilities to explore his identity as a writer and often expressed political points of view. In *Ghost of Chance*, becoming is the reaction to corporate greed and ignorance represented by Bradley Martin and the Board; and in many

ways, the search for personal freedom and tolerance has been Burroughs' project from the start.

**Funding:** This research received no external funding.

**Conflicts of Interest:** The author declares no conflict of interest.

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
