# Peer review of "Baboons, Centipedes, and Lemurs: Becoming-Animal from Queer to Ghost of Chance"

_humanities, doi:10.3390/h10010051_

Round 1
Reviewer 1 Report
This is a strong piece on Burroughs and becoming animal. Burroughs' oeuvre is handled with care and insight as are the theoretical borrowings from Deleuze and Guattari. I am generally compelled by the argument, although I would like a bit more of a conclusion. What is finally at stake with Burroughs' use of animal/human metamorphoses? What are the final politics at stake in the piece? A line of flight or is there a larger vision at stake? Also, the Burroughs criticism is voluminous (I have published on Naked Lunch) and citing a bit more of it is recommended. Harris is assuredly key, but there is also work by Timothy Murphy, Davis Schneiderman and many others. With proper citation and a bit more fleshed out of a conclusion, this piece is ready to be published.
Author Response
Thanks for the helpful comments. I added two paragraphs in the the conclusion to make clearer what the importance of becoming-animal is for Burroughs. Furthermore, I added references to a text by Adam Meyer about Kafka and Burroughs that I forgot about, because I read it ages ago. Also references to “Ccru” in Schneiderman, Timothy Murphy and the George Hart paper already published in this volume.
Reviewer 2 Report
Brief summary: The essay attempts - and succeeds - in examining the concept of 'becoming-animal' in Burroughs' work, and focuses on lesser-discussed texts in order to do so.
Broad comments: The opening section on Kafka takes some time to establish a link to Burroughs - would recommend more signposting of the trajectory of the discussion in the opening paragraph.
Although mentioning both Naked Lunch and centipedes, it may be worth fleshing this out a little more: perhaps the 'Meeting of International Conference of Technological Psychiatry' section of NL as an example of 'becoming' through transmogrification.
At the end of p. 5, the leap from Roosevelt After Inauguration to Ghost of Chance feels immense and unconnected.
In the Ghost of Chance section, it may be worth drawing together the connective threads, whereby Mission is a recycled character, as is Bradley Martin, and also to explain what The Articles outlined in the fourth paragraph are.
'In Search of Yage' - published originally in segments in magazines rather than as a book, and subsequently incorporated within The Yage Letters in 1963.
Structurally sound and presents a linear discussion with solid conclusion, although I'm not sure of the need for the last sentence, which feels like an add-on that's not been part of the discussion.
Specific comments:
L28 'that time' - what time?
L125-127 - would benefit from greater clarity here: Burroughs' writing is like Kafka's? His journey is? In what way?
L149 - equalled - equated?
L211 - explain offensive: deemed offensive by whom?
